# Potentiating antibiotic efficacy via perturbation of non-essential gene expression

Peter B. Otoupal[1,2,3,7], Kristen A. Eller [1,7], Keesha E. Erickson[1], Jocelyn Campos[1], Thomas R. Aunins[1] & Anushree Chatterjee [1,4,5,6✉]

Proliferation of multidrug-resistant (MDR) bacteria poses a threat to human health, requiring new strategies. Here we propose using fitness neutral gene expression perturbations to potentiate antibiotics. We systematically explored 270 gene knockout-antibiotic combinations in *Escherichia coli*, identifying 90 synergistic interactions. Identified gene targets were subsequently tested for antibiotic synergy on the transcriptomic level via multiplexed CRISPR-dCas9 and showed successful sensitization of *E. coli* without a separate fitness cost. These fitness neutral gene perturbations worked as co-therapies in reducing a *Salmonella enterica* intracellular infection in HeLa. Finally, these results informed the design of four antisense peptide nucleic acid (PNA) co-therapies, *csgD*, *fnr*, *recA* and *acrA*, against four MDR, clinically isolated bacteria. PNA combined with sub-minimal inhibitory concentrations of trimethoprim against two isolates of *Klebsiella pneumoniae* and *E. coli* showed three cases of re-sensitization with minimal fitness impacts. Our results highlight a promising approach for extending the utility of current antibiotics.

[1] Department of Chemical and Biological Engineering, University of Colorado at Boulder, Boulder, CO 80303, USA. [2] Lawrence Berkeley National Laboratory, Joint Bioenergy Institute, Emeryville, CA 94608, USA. [3] Biomass Science and Conversion Technology Department, Sandia National Laboratories, Livermore, CA 94551, USA. [4] Biomedical Engineering, University of Colorado at Boulder, Boulder, CO 80303, USA. [5] Sachi Bioworks, Inc, Boulder, CO 80301, USA. [6] Antimicrobial Regeneration Consortium, Boulder, CO 80301, USA. [7] These authors contributed equally: Peter B. Otoupal, Kristen A. Eller. ✉email: chatterjee@Colorado.EDU

Antibiotic resistance is one of the foremost problems facing humanity. Estimates for the yearly cost imposed by antibiotic resistance reaches as high as $55 billion in the United States[1] and €1.5 billion across Europe[2]. Both the World Economic Forum[3] and the World Health Organization[4] warn of calamitous economic and health outcomes if current trends continue unabated; while approximately 1 million people die from such infectious yearly, annual deaths attributable to antibiotic resistant infections are estimated to reach 10 million by 2050[5]. This problem is likely to be exacerbated as multidrug-resistant (MDR) bacteria continue to emerge, necessitating the pursuit of alternative antimicrobial strategies.

Current antibiotic research has focused largely on developing new drugs that exhibit bactericidal activity through novel mechanisms. This has led to what is often referred to as the antibiotic arms race[6,7]. In this model, while scientists continue to discover and bring to market novel drugs, it is almost universally accepted that bacteria will continue to evolve resistance and thus inexorably diminish a therapy's efficacy over time[8]. The remarkable capacity for microbes to evolve resistance is due to a combination of relatively quick doubling times and large population sizes, allowing bacteria to rapidly explore a diverse array of genetic possibilities and increasing the likelihood for a fortuitously beneficial variant to emerge. Without addressing this central law governing antibiotic resistance, the antibiotic arms race seems all but perpetual.

Here we seek to highlight a potential therapeutic approach to tackling this fundamental problem in combating antibiotic resistance. Rather than focusing on the development of novel drugs utilizing alternative mechanisms for killing bacteria, this approach aims to devise co-therapies whose sole function is to potentiate the efficacy of existing antibiotic treatment. Crucially, such therapies must be designed to have minimal direct impact on bacterial fitness when administered independently. In doing so, these treatments should theoretically minimize inherent selective pressure for bacteria to evolve escape strategies that negate their impact.

The design of such therapies has been made possible with recent advances in synthetic biology and the corresponding development of novel tools with sequence-specific targeting capabilities. This includes transcriptome editing based on clustered regularly interspaced short palindromic repeats (CRISPR)-associated endonucleases (such as Cas9), which has been employed as an antimicrobial to selectively degrade particular genetic elements[9,10]. CRISPR has also been employed to interfere with the expression of target genes (CRISPRi) and has been explored for antibacterial applications[11]. Another tool for sequence-specific gene targeting are peptide nucleic acids (PNAs), single stranded DNA mimics that bind tightly to the corresponding antisense mRNAs[12]. PNAs have demonstrated effective antimicrobial activity when conjugated to cell-penetrating-peptide (CPP) motifs and targeted to knockdown gene expression[13–17]. Either of these approaches could be used to reduce expression of particular genes to enhance antibiotic killing.

To design fitness-neutral potentiating therapies based on these technologies, we must first understand what genes to target. To this end, we began with a systematic exploration of gene-drug synergy in the well characterized bacteria *Escherichia coli*, for which there exists a collection of every viable gene knockout[18]. Successful growth in the absence of these genes suggests that their loss provides a minimal fitness impact, fulfilling our first criteria of designing therapies that themselves are harmless to bacteria. Their existence also enables rapid screening of how analogous gene knockdowns will impact antibiotic efficacy. We focused on thirty stress-response gene knockouts that we identified to be differentially expressed during antibiotic exposure in our previous works[19,20]. We systematically characterized how each of the strains responded to growth in sub-minimum inhibitory concentrations (MICs) of nine commonly used antibiotics representing a broad spectrum of functional classes. From this we identified significant synergistic interactions and constructed CRISPRi constructs to replicate their results individually and in a multiplexed fashion. These constructs were also applied to *Salmonella* infections of HeLa cells to demonstrate their therapeutic potential in amplifying antibiotic efficacy. Finally, we utilize PNAs to knockdown these genes in four clinically isolated, MDR strains of bacteria.

Collectively, this study demonstrates a novel pipeline of synergistic drug discovery based on the perturbation of non-essential genes, progressing from knockouts, to perturbations, to infection models, and ultimately to PNA therapies. We show that this approach successfully re-sensitizes the bacteria to antibiotic treatment without imposing its own fitness cost. Together, these results demonstrate a new platform for discovering and designing synergistic therapies for enhanced antimicrobial treatment.

**Table 1 The 30 genes in *Escherichia coli* BW25113 investigated in this study.**

| Gene | Description | Function |
|------|-------------|----------|
| *polB* | DNA polymerase II | DNA processes |
| *recA* | DNA strand exchange and recombination protein | |
| *dam* | DNA adenine methyltransferase | |
| *dinB* | DNA polymerase IV | |
| *mutS* | Methyl-directed mismatch repair | |
| *wzc* | Colanic acid biosynthesis protein | Metabolism |
| *tar* | Methyl accepting chemotaxis protein | Motility |
| *flhC* | FlhC-FlhD transcriptional regulator of flagellum biogenesis | |
| *flhD* | FlhC-FlhD transcriptional regulator of flagellum biogenesis | |
| *ydiV* | Anti-FlhDC factor | |
| *bglG* | Uptake and utilization of β-glucosides | Regulation |
| *crp* | cAMP receptor protein, regulates energy metabolism | |
| *csgD* | Regulates curlin genes, important for biofilm formation | |
| *fur* | Ferric uptake regulator | |
| *gadX* | Controls transcription of pH-inducible genes | |
| *hfq* | RNA-binding protein | |
| *phoP* | Two component regulatory system phoQ/phoP | |
| *rob* | Transcriptional regulator induced by dipyridyl, bile salts, or decanoate | |
| *rpoS* | RNA polymerase, sigma S | |
| *fnr* | Regulator, mediates aerobic to anaerobic transition | |
| *marA* | Multiple antibiotic resistance regulator | Redox |
| *soxS* | Regulation of superoxide response regulon | |
| *ydhY* | Putative oxidoreductase system protein | |
| *ybjG* | Putative bacitracin resistance protein | |
| *cyoA* | Cytochrome bo terminal oxidase subunit II | Transport |
| *tolC* | AcrAB-TolC multidrug efflux pump—membrane fusion protein | |
| *acrA* | AcrAB-TolC multidrug efflux pump—membrane fusion protein | |
| *fiu* | Outer membrane receptor for iron transport | |
| *yjjZ* | Unknown | Unknown |
| *yehS* | Unknown | |

**Table 2 The nine antibiotics investigated in this study and their working concentrations.**

| Antibiotic | Abbreviation | Dose (μg/mL) | Target |
|---|---|---|---|
| Ampicillin | AMP | 2 | Cell wall |
| Ceftriaxone | CRO | 2 | Cell wall |
| Tetracycline | TET | 0.25 | Protein synthesis (30S) |
| Erythromycin | ERY | 50 | Protein synthesis (50S) |
| Chloramphenicol | CM | 0.4 | Protein synthesis (50S) |
| Puromycin | PURO | 50 | Protein synthesis (RNA) |
| Ciprofloxacin | CIP | 0.008 | DNA replication |
| Sulfadimidine | SDI | 0.5 | DNA/RNA synthesis |
| Trimethoprim | TMP | 0.125 | DNA/RNA synthesis |

## Results

**Selecting gene targets and antibiotic concentrations.** We utilized the ample evidence in literature of genes associated with stress response and/or adaptation processes to narrow down a set of promising candidates for potentiating antibiotic activity[21]. Many studies have investigated *E. coli* response to various antibiotics, including the use of whole genome library knockouts[22–24]. Of the 4000+ genes in *E. coli*, these studies have identified a set of ~100-300 genes that influence the bacteria's sensitivity to multiple antibiotics. This includes the *tolC* and *acrA* genes comprising the TolC-AcrAB efflux pump of *E. coli*, which has been frequently explored as a hub for targeting antibiotic resistance in these and many other studies[25–29]. Similarly, the transcriptional regulator *marA*, *soxS*, and *rob* have each been implicated in increasing *E. coli* antibiotic resistance[30–32]. Mutants of genes involved in the SOS response such as *recA*, *dinB*, *polB*, and *dam* are known to strongly influence *E. coli* response to antibiotics[33–38]. Additionally, a set of genes have been show to exhibit increased transcriptional activity during exposure to stressful environments in general, including *rpoS*, *mutS*, *hfq*, and *cyoA*[39–44].

We have previously performed our own transcriptional analysis of bacteria adapting to stress response. Several genes were found to impact adaptive resistance in a transcriptome-level analysis of adapted versus unadapted strains (*fiu*, *tar*, *wzc*, *yjjZ* were differentially expressed while *ybjG*, *ydhY*, *ydiV*, and *yehS* were differentially variable)[19,20]. In these studies, we also investigated the transcriptional regulators that control the genes exhibiting differential expression and narrowed down a set of novel targets (including *bglG*, *crp*, *csgD*, *flhC*, *flhD*, *fnr*, *fur*, *gadX*, and *phoP*). We reasoned that because these genes (or the downstream genes they regulate) exhibited particularly high gene expression variability during adaptation, they would serve as interesting candidates for targeting expression interference therapies towards.

Combining these two approaches, we developed a final set of thirty selected genes to explore as potential targets for antibiotic combination therapy (Table 1). These genes represent diverse functionalities, including transport (*acrA*, *tolC*, and *fiu*), mutagenesis (*mutS*, *dam*, *polB*, *dinB*), motility (*tar*, *flhC*, *flhD*, and *ydiV*), uncharacterized function (*yjjZ* and *yehS*), or general global regulation (the aforementioned others). Knockouts of gene were obtained from the Keio collection[18]. As *E. coli* is viable upon deletion of these genes, we reasoned that reducing expression of these non-essential genes is less likely to impose an inherent fitness cost compared to reducing expression of essential genes.

We chose to test these knockouts' growth in a set of nine antibiotics representing a diverse set of common antimicrobial therapies (Table 2). Growth assays were first performed to determine a suitable dosage for each antibiotic agent below the minimum-inhibitory concentration (MIC) (Fig. S1). Antibiotic concentrations resulting in a 10–50% fold reduction in growth were identified and used for all subsequent testing[45].

**Gene knockout synergy with antibiotic treatment.** We characterized BW25113 growth in 270 combinations of 30 Keio knockouts with these nine antibiotics. Synergy (S) between gene knockout and antibiotics was determined using the Bliss Independence model, $S = W_X * W_Y - W_{XY}$, based on bacterial fitness in the presence of antibiotic ($W_X$), gene knockout ($W_Y$), or a combination of both ($W_{XY}$). An example of this is presented in Fig. 1a. Here, ampicillin (AMP) exhibited no significant impact on BW25113's fitness ($W_X = 1.04 \pm 0.06$), nor did deletion of *acrA* ($W_Y = 0.93 \pm 0.03$). However, BW25113-Δ*acrA* exposed to AMP demonstrated poor fitness ($W_{XY} = 0.07 \pm 0.02$), indicating potentiation of AMP activity by removing *acrA* ($S = 0.89 \pm 0.12$). This process was repeated for all 269-remaining gene–drug pairs (Fig. 1b). Drug independent (additive) and drug dependent (synergistic or antagonistic) interactions were classified using a two-sided *t*-test, after log transformation of the data, as described by Demidenko et al.[46]. If the interaction was statistically significant ($P < 0.05$) then a one-sided *t*-test further determined synergism ($P < 0.05$) or antagonism ($P > 0.05$).

We identified 90 gene–drug interactions which elicited synergistic interactions, and another 67 gene–drug interactions that resulted in antibiotic antagonism (Fig. 1b). Of the thirty genes explored, five (*acrA*, *fur*, *recA*, *rpoS*, and *tolC*) were found to potentiate activity of at least six out of nine antibiotics tested, suggesting that these genes are promising targets for co-therapies. Many of these gene knockout drug synergies, to the best of our knowledge, are novel discoveries, such as that between *fnr* and five of the antibiotics tested. Conversely, two gene knockouts (*gadX* and *hfq*) caused antagonism of at least seven of nine antibiotics and are thus poor candidates for antimicrobial potentiation purposes.

The goal of this systematic investigation was two-fold, the first of which was to identify strong gene–drug synergistic interactions. The second goal was to ensure that these knockouts imposed minimal fitness costs on cell growth, to protect against the possibility of natural selection working against sequence-specific therapies designed to target these genes. In order to avert development of gene knockdown therapies that pose a direct impact on cellular fitness in the absence of antibiotic exposure, we avoided gene knockouts that had particularly strong impacts on growth. This includes four gene deletions: Δ*dam* ($W_Y = 0.49 \pm 0.28$), Δ*rob* ($W_Y = 0.48 \pm 0.05$), Δ*hfq* ($W_Y = 0.38 \pm 0.04$), and Δ*tar* ($W_Y = 0.41 \pm 0.05$). Δ*hfq* and Δ*gadX* demonstrated consistent antagonism and were thus already not candidates for further screening.

Collectively, these results point to promising targets for designing fitness-neutral gene expression perturbations that enhance antibiotic efficacy. We explore the development of such therapies utilizing CRISPR-Cas9 in a latter portion of the manuscript. However, this knockout-drug synergy screen also provides interesting conclusions regarding the general mechanisms of gene–drug synergy that we will now explore.

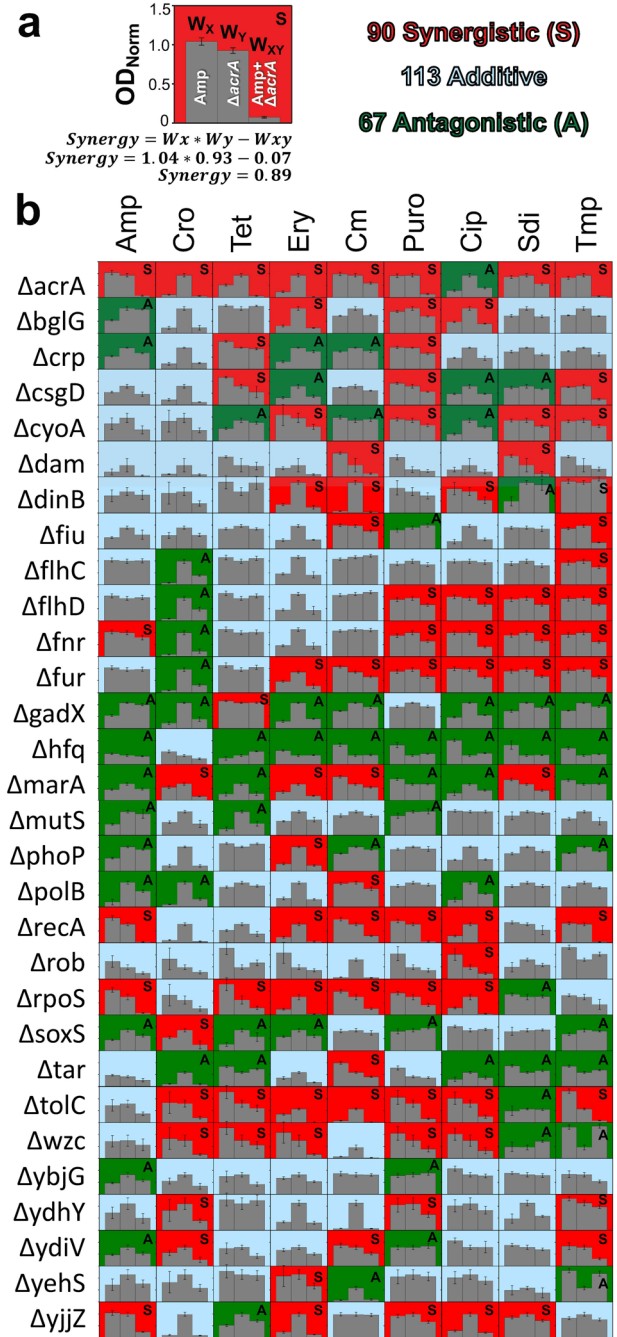

**Fig. 1 Synergistic interactions between *E. coli* BW25113 gene knockouts and drug treatment. a** An example of how synergy values were calculated. Strain fitness (*W*) was calculated as the maximum optical density reached during 16 h of growth, normalized to the maximum optical density of wildtype BW25113 with no antibiotic exposure during the same 16 h growth period. Fitness was calculated in the presence of antibiotic ($W_X$), gene knockout ($W_Y$), or antibiotic treatment of a gene knockout strain ($W_{XY}$). Synergy (*S*) was calculated as $W_X * W_Y - W_{XY}$, with positive values indicating synergy and negative values indicating antagonism. This example shows that deletion of *acrA* potentiates antibiotic activity of ampicillin against BW25113. **b** This process was performed for all 270 gene–drug combinations. Interactions that proved significantly synergistic (or antagonistic) are color-coded red and have an "S" (or green and an "A"). Non-significant interactions are classed as additive (blue and contain no letter distinction). All bar graphs' *y*-axes use the same scale (from 0.0 to 1.5) used in Fig. 1a. Error bars represent standard deviation of at least three biological replicates.

**Unraveling mechanistic insights from knockout-drug synergy.** We first explored the applicability of a hypothesis raised in our previous work[47] stating that the degree of gene–drug synergy is influenced by the epistatic interactions the targeted gene is involved with. In particular, we have found that as more protein–protein interactions are disrupted by targeted genetic perturbations, a greater than expected detrimental fitness impact emerges. We applied a similar approach in this study to explore the known protein interaction networks of each of the thirty genes here to unravel similar underlying correlations. From the STRING database[48], we extracted information of all the known and predicted protein–protein interactions for each gene to construct and quantify their protein interaction networks. We explored the total number of proteins interactions, i.e., the amount of nodes that are present (Fig. S2). We found that significantly greater synergy was observed with increased number of proteins interacting (*P* = 0.006). While correlation in itself does not prove a meaningful connection, its existence suggests that targeting nodes involved in broad protein interaction networks can lead to enhanced levels of gene–drug synergy. Additionally, two other studies from our lab have also found a similarly significant correlation between synergy and protein interactions[17,47]. Our previous work demonstrated that perturbing genes involved in broad protein interaction networks lead to strong negative epistatic effects. The data presented here supports this observation and lends further credence to the notion that epistasis plays a significant role in influencing bacterial fitness response towards co-therapies.

To further explore the mechanistic underpinnings of gene–drug interactions, antibiotics and genes were grouped into mechanisms of action and pathways respectively (Fig. 2). Notably, the one knockout directly affecting metabolism, Δ*wzc*, represented one of the top three synergistic knockouts in ceftriaxone (CRO), erythromycin (ERY), and ciprofloxacin (CIP), but was also one of the three most antagonistic knockouts in sulfadimidine (SDI) and trimethoprim (TMP). Whole genome RNA-sequencing showed that *wzc*, a colanic acid biosynthesis gene, was overexpressed during AMP exposure[20], although no significant synergy was observed between Δ*wzc* and AMP in this experiment. Though the classes of antibiotics in which synergy was observed were diverse, clear antagonism emerged in the antibiotics related to DNA/RNA synthesis (sulfadimidine and trimethoprim).

The TolC-AcrA efflux pump knockouts demonstrated some of the highest levels of gene–drug synergy. Knockouts of at least one of these genes was always one of the three most synergistic genetic changes for all the antibiotics tested, apart from the 50S targeting antibiotics ERY and chloramphenicol (CM). However, even in these antibiotics, both knockouts resulted in significant synergy. The remaining standout knockouts include Δ*recA*, Δ*dam*, and Δ*rpoS*, which demonstrated high synergy with four, two, and two antibiotics, respectively. There was no clear relationship between the cellular processes these genes are involved in and the antibiotics' modes of action.

**Introducing gene–drug synergy using CRISPRi.** If knocking out these genes resulted in significantly amplifying antibiotic potency, we hypothesized that lowering their expression without completely removing them from the genome might engender similar results while also demonstrating a more therapeutically viable application of gene–drug synergy. To this end, we developed CRISPR interference (CRISPRi) constructs to knockdown expression of genes showing significant antibiotic synergy. For this, catalytically dead Cas9 (dCas9) was employed to reduce mRNA production of the targeted gene, which was subsequently exposed to a variety of antibiotics to determine its potential for inducing gene–drug synergy (Fig. 3a).

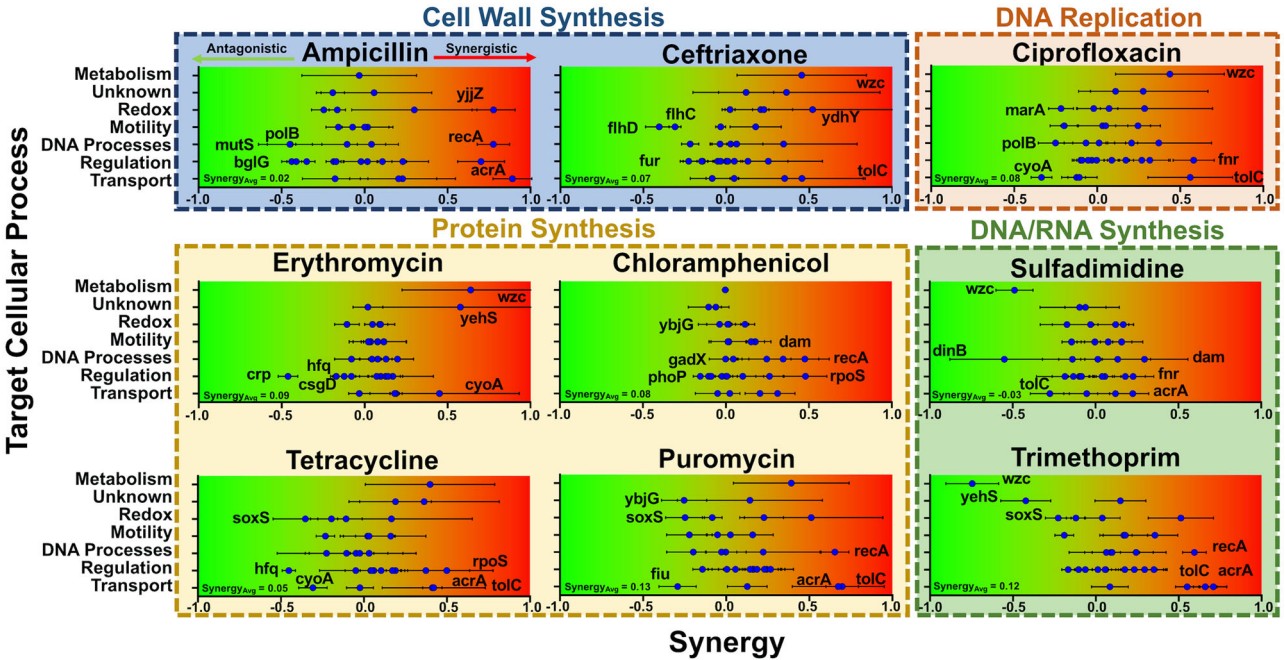

**Fig. 2 Correlations in gene–drug synergy.** Degree of synergy between gene knockouts and antibiotic treatments, grouped by biological mechanisms. Gene knockouts are separated into their specific cellular processes on the *y*-axis, with corresponding synergy plotted on the *x*-axis, going from antagonistic (green, left) to synergistic (red, right). Antibiotics are further grouped based on the mechanism of action, such as targeting cell wall synthesis. The top three synergistic interactions and top three antagonistic interactions are specifically labeled in each graph. In the bottom left of each graph is listed the average synergy of all thirty gene knockouts with the specific antibiotic. Error bars represent standard deviation of at least three biological replicates propagated from fitness values.

We specifically focused on six of the genes showing the greatest degree of synergy with each of the antibiotics tested, while also maximizing the diversity of genetic pathways targeted for synergy with each antibiotic (Table S1). We utilized a dual-plasmid system based on the original CRISPRi system to deliver gene knockdown constructs to BW25113[49]. One plasmid encoded expression of dCas9 under the anhydrous tetracycline (aTc)-inducible promoter, while the other encoded a unique single guide RNA (sgRNA). sgRNA targets were designed using criteria set forth in previous studies for successfully eliciting inhibition[13,50]. This includes targeting either within the first ~50 nucleotides of the gene's open reading frame, or within the −35 to +1 site of the gene's promoter sequence. The ability of six of these constructs (*marA*-i, *recA*-i, *acrA*-i, *tolC*-i, *soxS*-i, and *wzc*-i) and other similarly designed constructs to inhibit gene expression have been verified in our previous studies using real-time quantitative PCR[11,19,47]. We also analyzed all potential off-targets for each of these constructs based on what constitutes most likely off-targets in *E. coli*, and outline each of them in Table S2[51,52]. Of these, only four constructs (*soxS*-i, *tolC*-i, *ydhY*-i, and *phoP*-i) had potential off-targets of genes whose deletions are known to cause fitness defects. As the two plasmids used to express the CRISPRi constructs rely on AMP and CM selection markers, we excluded exploration of these antibiotics going forward. Additionally, due to the general low degree of synergy demonstrated by gene knockouts with SDI, this antibiotic was not included.

The gene-antibiotic synergy experiments were again repeated, with CRISPRi employed in the place of gene knockouts. A strain expressing a sgRNA targeting the coding sequence of red fluorescent protein (*rfp*) (which is not present in the strain) was used in lieu of wildtype BW25113 as the control. All fitness values were normalized to the growth of the control strain in the absence of any antibiotic. Fitness of the control strain was measured during exposure to each antibiotic ($W_X$), each individual perturbation strain in the absence of antibiotic ($W_Y$), and each perturbation strain during antibiotic exposure ($W_{XY}$). Experiments were again performed in M9 minimal media supplemented with 0.4% glucose and antibiotics as appropriate.

Notably, the majority of our CRISPRi constructs appeared to slightly improve growth over the *rfp* targeting control in the absence of antibiotics, as indicated by $W_Y$ values greater than 1.0. This phenomenon likely stems from a fundamental slight growth impact caused by the *rfp* targeting construct. One plausible explanation of this is that the *rfp* targeting construct was more prone to off-targeting effects due to the lack of an on-target sink to draw the dCas9-sgRNA complex towards. A recent report systematically examined the off-targeting potential of dCas9 in *E. coli* and found that binding of as little as five bases in the seed sequence is sufficient to have a measurable impact on transcription[52]. We performed a systematic exploration of the most likely off targets for this *rfp*-i control construct (Table S2). The most likely off-targets with potential fitness impacts include *cysG*, *ftsI*, and *metL*. We also investigated the off-targets of our other CRISPRi constructs. However, as each of these constructs has an actual on-target, and none presented a $W_Y$ lower than 1.0, the influence of off-targets is likely trivial. These results suggest that a better control for future studies utilizing CRISPRi would utilize a DNA sequence present in the genome but known to be uninvolved with fitness.

Despite this caveat for our control, the majority of CRISPRi perturbation co-therapies resulted in statistically significant potentiation of antibiotic treatment, suggesting that this was an effective strategy for replicating revealed knockout-drug synergy (Fig. 3b). Four perturbations improved efficacy of ERY (*acrA*-i, *tolC*-i, *cyoA*-i, and *wzc*-i), three synergized with TMP (*acrA*-i, *tolC*-i, and *recA*-i) and CIP (*rpoS*-i, *recA*-i, and *fnr*-i), and two synergized with puromycin (PURO) (*acrA*-i and *tolC*-i) and

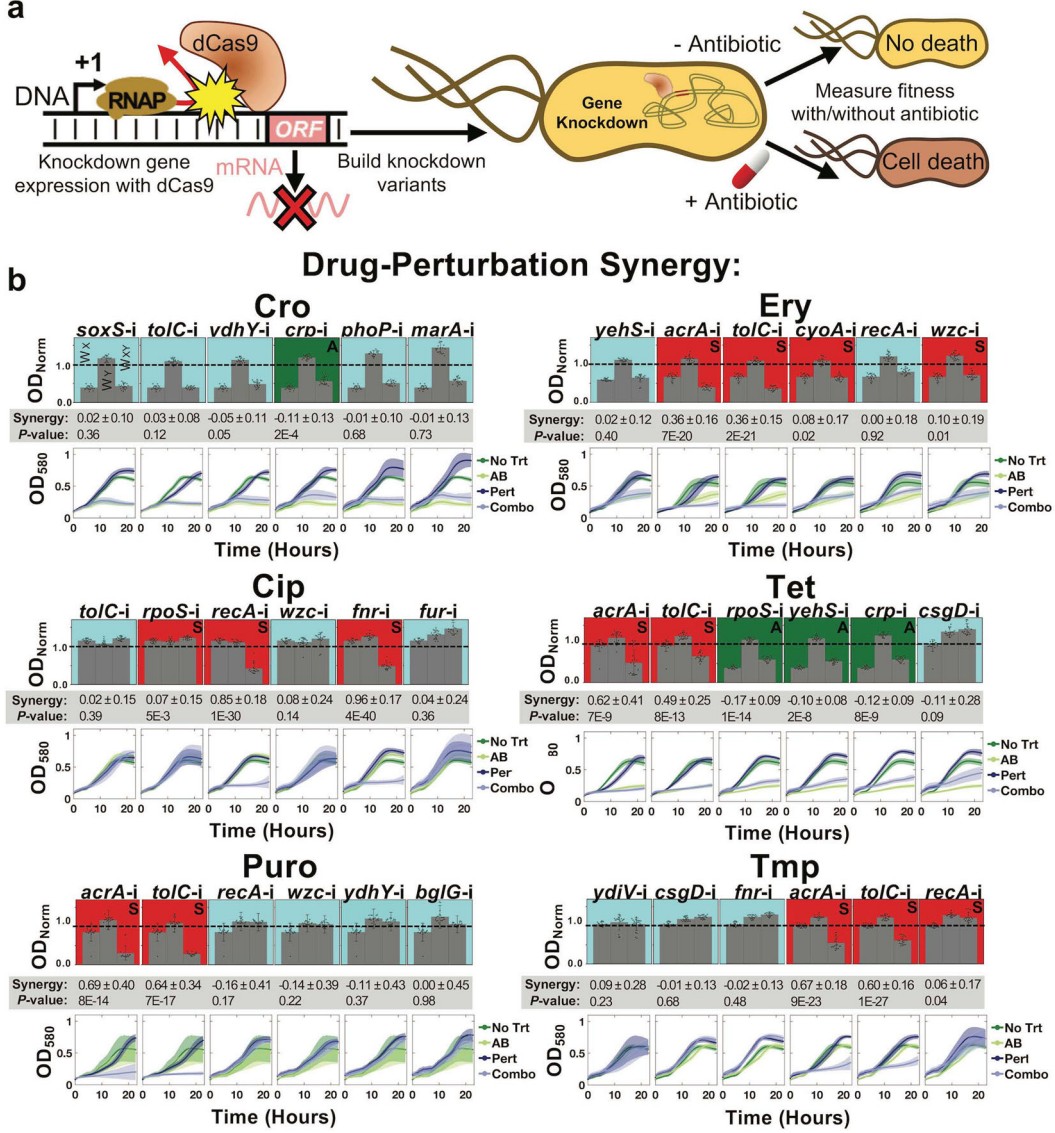

**Fig. 3 Applying CRISPRi to potentiate antibiotic treatment. a** dCas9 is targeted to promoter or open reading frame elements of specific genes, preventing RNA polymerase from transcribing DNA into mRNA. Constructs were created to block transcription of six genes for which deletion resulted in significant synergy with a specific antibiotic. **b** Each of these CRISPRi strains were tested for their synergy with antibiotic treatment. Strain ODs' after 16 h of growth in M9 minimal media were quantified and these values were used to calculate fitness. The growth of the control *rfp* perturbation strain during exposure to the listed antibiotic, the growth of the gene perturbation with no antibiotic, and the growth of the gene perturbation in the presence of the listed antibiotic were all normalized to the growth of the control strain without exposure to antibiotic, giving $W_X$, $W_Y$, and $W_{XY}$ respectively. Statistically significant synergy or antagonism is indicated by a red background and an "S" or green background and an "A" respectively, with additive interactions shown in blue. Synergy values are listed below each graph with their associated significance. Error bars represent standard deviation of at least 20 biological replicates. Growth curves of each associated bar graph are shown below. Dark green lines indicate the control, light green lines indicate antibiotic only exposure, dark blue lines indicate CRISPRi perturbation (Pert in legend), and light blue lines indicate combination. Error bars represent standard deviation of at least 20 biological replicates and gray circles represent individual biological replicates.

tetracycline (TET) (*acrA*-i and *tolC*-i). Three combinations with TET resulted in clear antagonism: *rpoS*-i, *yehS*-i, and *crp*-i. A few CRISPR perturbations did stand out from the rest in the clear antibiotic synergy they induced. Most notable is the degree of synergy induced by inhibitions of the *tolC-acrA* efflux pump in ERY (*acrA*-i = 0.36 ± 0.16, *tolC*-i = 0.36 ± 0.15), PURO (*acrA*-i = 0.69 ± 0.40, *tolC*-i = 0.64 ± 0.34), TET (*acrA*-i = 0.62 ± 0.41, *tolC*-i = 0.49 ± 0.25), and TMP (*acrA*-i = 0.67 ± 0.18, *tolC*-i = 0.60 ± 0.16). Inhibitions of *recA* and *fnr* additionally showed significant improvements in CIP efficacy (S = 0.85 ± 0.18 and 0.96 ± 0.17, respectively). CRISPRi largely replicated gene knockout synergy with antibiotic treatment. A direct comparison of the

synergy levels is presented in Fig. S3. A few perturbations produced greater synergy in the perturbation context than in the knockout context, including both *acrA*-i and *tolC*-i combined with ERY treatment, and both *fnr*-i and *recA*-i combined with CIP treatment. Comparing synergy values between gene knockouts and CRISPRi gene perturbations showed a strong correlation (P = 0.003) Additionally, the average synergy across all perturbations was lower than the average synergy of corresponding gene knockouts during exposure to CRO (CRISPRi = −0.02, Δ = 0.23), TET (CRISPRi = 0.10, Δ = 0.38), ERY (CRISPRi = 0.15, Δ = 0.36), PURO (CRISPRi = 0.16, Δ = 0.53), and TMP (CRISPRi = 0.23, Δ = 0.49). Lower levels of perturbation-antibiotic

synergy relative to knockout-antibiotic synergy are unsurprising given that the target gene can still be expressed (albeit at a lower level) in the former case.

To have a better understanding of these perturbations on antibiotic efficacy, we also performed growth curve analysis of each CRISPRi strain with their corresponding antibiotics in an alternative environment of LB media (Figs. S4–9). In most cases, synergy between CRISPRi and antibiotics is made apparent in the early stages (5–10 h) of growth. Significant growth inhibition was caused by CRISPRi synergism with the associated antibiotic in the following conditions: four with CRO (*tolC*-i, *ydhY*-i, *phoP*-i, and *marA*-i), five with CIP (*tolC*-i, *recA*-i, *rpoS*-i, *wzc*-i, and *fur*-i), four with TET (*tolC*-i, *rpoS*-i, *crp*-i, and *csgD*-i), two when targeting ERY (*tolC*-i and *acrA*-i), three when targeting TMP (*tolC*-i, *acrA*-i, and *ydiV*-i), and three when targeting PURO (*tolC*-i, *recA*-i, and *acrA*-i). While the majority of perturbations induced antibiotic synergy in at least one of the two environments tested, a few perturbations provided either no synergism or was antagonistic and are therefore not useful candidates for potential therapies. This includes targeting *crp* during CRO treatment, *yehS* during TET treatment, and both *wzc* and *blgG* during PURO treatment. These data sets provide quantification of gene-drug synergies in two environments. Taken together, these results highlight that CRISPRi can effectively potentate antibiotic treatment.

**Multiplexing CRISPRi exacerbates antibiotic synergy.** An advantage of CRISPRi is the relative ease in which individual perturbations can be combined into a single cell by including multiple sgRNAs in a CRISPR array. Furthermore, we have previously shown that multiplexing perturbations tends to exacerbate detrimental fitness impacts by inducing negative epistatic interactions between the perturbed genes[47]. This suggests that multiplexing synergistic CRISPRi perturbations could exacerbate the potentiation of antibiotic efficacy. We took advantage of this by combining the six perturbations designed for each antibiotic into one construct and testing their impacts on BW25113 growth during antibiotic exposure.

We first ensured that expanding the number of perturbations did not have an inherent impact on growth by testing the growth of a control strain harboring six tandem *rfp* perturbations (Fig. 4a). This strain exhibited no significant shift in basal fitness, nor did it show antagonism or synergy with any antibiotic.

In stark contrast to this, every multiplexed CRISPRi perturbation strains designed for inducing synergy showed significant potentiation of antibiotic efficacy, apart from PURO perturbations (Fig. 4a). Synergy was particularly pronounced with TMP ($0.25 \pm 0.14$, $P = 4E{-}12$), CIP ($0.22 \pm 0.48$, $P = 0.01$), and ERY ($0.20 \pm 0.16$, $P = 3E{-}8$). The strong synergy observed by multiplexed TET perturbations ($0.13 \pm 0.09$, $P = 7E{-}10$) is particularly notable given the varied levels of synergy observed when perturbations were applied individually.

To further elucidate multiplexed perturbations' impacts on BW25113 growth, we examined each strain's growth profile over 20 h in both the presence and absence of antibiotic and compared these profiles to the multiplexed control perturbation strain (Fig. 4b). All strains grew identically to the control in the absence of antibiotic exposure, apart from a slight lag-time shift in the multiplexed perturbation under TET treatment. In the presence of antibiotics, multiplexed perturbation strains demonstrated diminished growth capacity compared to the control strain, apart from multiplexed PURO perturbations. For PURO multiplexed perturbations, one possibility for why multiplexing was less effective than individual gene targeting is the dilution of dCas9 protein when targeting multiple genomic loci simultaneously,

leading to less gene repression. We have previously demonstrated that multiplexed gene perturbations resulted in similar levels of gene repression using individual gene perturbations[11,47]. Furthermore, this potential problem would be alleviated in a therapeutic context by utilizing direct delivery of dCas9-sgRNA pre-assembled complexes. Regardless, these results support the conclusion that the designed multiplexed perturbations largely potentiate antibiotic treatment without imposing a substantial, direct impact on fitness, and provided direction as to which perturbation-antibiotic pairs to focus on for the remainder of our experiments.

To further confirm fitness impacts of multiplexed perturbations, we employed a competition assay on the two multiplexed perturbations exhibiting the greatest degree of synergy and strongest impact on growth: the strains designed for synergizing with TET and TMP. In this assay, perturbed strains were co-cultured with the control strain, and the relative abundance of each strain was determined at the beginning and end of competition. For this, the control CRISPRi strain harboring *rfp* perturbation was modified to constitutively express mCherry. The relative abundance of each multiplexed perturbed strain was equivalent to that of the control strain in the absence of antibiotic exposure, confirming that these perturbations have no direct impact on fitness (Fig. 4c, d). The results noticeably changed when antibiotics were included; the relative abundance of each multiplexed perturbed strain dropped significantly, indicating a substantial impact on bacterial fitness. Synergy was calculated using the same equation as before, replacing ODs with direct measurements of viable colony-forming units (CFUs). This revealed statistically significant potentiation of antibiotic efficacy ($W_{XY} = 0.64 \pm 0.22$, $P = 1E{-}4$, and $W_{XY} = 0.61 \pm 0.10$, $P = 1E{-}4$ for TET and TMP multiplexed perturbations, respectively) (Fig. 4c, d). Collectively, these results demonstrate that multiplexed perturbations further potentiate antibiotic efficacy while minimizing direct fitness impacts.

**CRISPRi potentiates antibiotic efficacy in infection models.** A benefit of the CRISPRi strategy for enacting sequence-specific gene therapies is the relative ease with which it can be applied to a vast array of organisms. For instance, many of these CRISPRi constructs can be directly applied to a pathogenic relative of *E. coli*, the bacteria *Salmonella enterica* (Fig. 5a). An analysis of the genome of *S. enterica* serovar Typhimurium SL1344 shows that six sgRNAs designed for BW25113 have complete (*acrA*, *cyoA*, and *fnr*) or near-complete (*crp*, *rpoS*, and *tolC*) homology to SL1344's genome, and therefore are likely to maintain efficacy in this organism (Fig. 5b). SL1344 is a model organism for studying bacteria in intracellular infections due to the relative ease in which it infects human cell lines[53]. To explore the potential for gene expression perturbations to potentiate antibiotic efficacy in a therapeutic context, we created two new sgRNA plasmids: one harboring all six targets, and another harboring just the three with perfect homology. These plasmids, as well as the single *rfp*-targeting control plasmid, were co-transformed into SL1344 with the Cas9 expression plasmid, and antibiotic synergy was again explored.

No detrimental impact on basal SL1344 fitness was observed in the absence of antibiotics for either of these strains ($W_Y = 1.11 \pm 0.18$ and $W_Y = 0.97 \pm 0.07$ for the three and six perturbations respectively) (Fig. 5c, d). Significant synergy was again observed in several instances. Both multiplexed perturbed strains showed significant synergy with CRO and TMP, the latter of which appeared to be particularly potentiated ($S = 0.12 \pm 0.30$, $P = 0.01$ and $0.21 \pm 0.15$, $P = 3E{-}10$ for three and six perturbations respectively). Strong synergy was also observed between ERY and

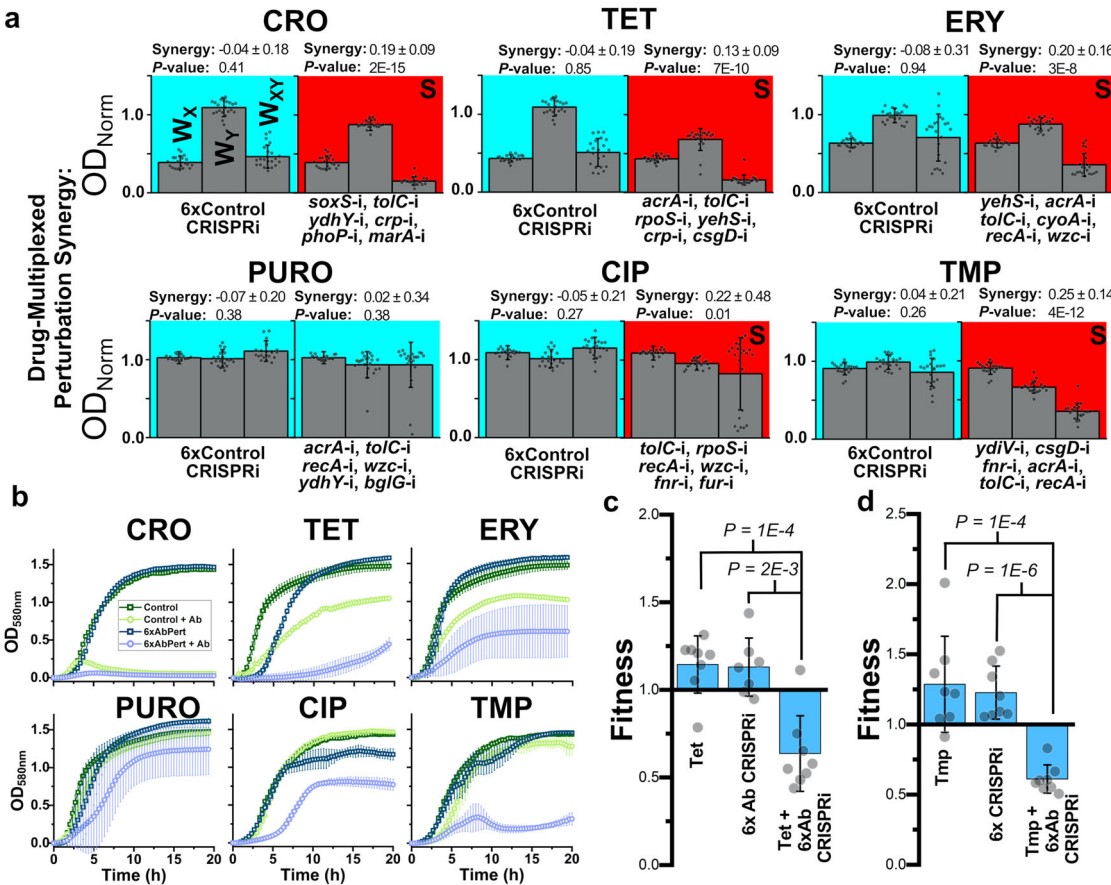

**Fig. 4 Multiplexed CRISPRi perturbations further potentiate antibiotic treatment. a** The six individual gene perturbations designed for each antibiotic were multiplexed into one strain, and synergy was again screened for (right column). A control strain with six nonsense *rfp* perturbations was also created to show that harboring multiple targets did not influence these results (left column). Strain ODs' after 16 h of growth were quantified, and these values were used to calculate fitness. For the left column under each antibiotic, growth of the control single *rfp* perturbation strain during exposure to the listed antibiotic, growth of the six *rfp* perturbation strain with no antibiotic, growth of the six *rfp* perturbation strain in the presence of the listed antibiotic were all normalized to the growth of the single *rfp* control strain without exposure to antibiotic, giving $W_X$, $W_Y$, and $W_{XY}$, respectively. The same occurred on the right column, except the control single *rfp* perturbation strain was replaced with the six *rfp* perturbation strain, and the six *rfp* perturbation strain was replaced with the six multiplexed gene perturbation strains designed for each antibiotic. Statistically significant synergy is indicated by a red background and the letter "S" and additive by a blue background. Error bars represent standard deviation of 22 biological replicates and gray circles represent individual biological replicates. Synergy values are listed above each graph with significance. **b** Growth curves of these multiplexed CRISPRi strains in the presence of each antibiotic. Error bars represent standard deviation of three biological replicates. A more thorough fitness assay using competition was applied to more precisely estimate the fitness impacts of multiplexed perturbations for **c** TET and **d** TMP. Competition was performed for these strains against a fluorescent control strain harboring one nonsense CRISPRi perturbation in either the presence or absence of antibiotic treatment. A control competition of the 6× *rfp* perturbation strain against the fluorescent control was also performed in the presence of antibiotic. Fitness was calculated using the standard Malthusian fitness equation (see "Methods" section). Error bars represent standard deviation of eight biological replicates.

the six-perturbation strain ($0.29 \pm 0.13$, $P = 1E-8$). Additionally, antagonism was observed between the three and six perturbations and CIP ($-0.15 \pm 0.31$, $P = 0.01$ and $-0.07 \pm 0.13$, $P = 5E-3$, respectively), which could be due to the antagonism demonstrated by $\Delta acrA$ and $\Delta cyoA$ in BW25113. Antagonism was also seen in the three-perturbation strain combined with ERY ($-0.21 \pm 0.31$, $P = 3E-3$).

Going forward, we focused our efforts on characterizing these perturbations' impacts on TET and TMP, as these antibiotics showed high levels of synergy. The growth profiles of each strain were characterized in the presence of no antibiotic, TET, or TMP (Fig. 5e). No impact on growth was observed in the absence of antibiotic exposure, while detrimental impacts were observed for the perturbed strains during antibiotic exposure. This again indicates that perturbations imposed no direct fitness cost while still potentiating antibiotic treatment of SL1344.

To investigate the ability of perturbations to potentiate antibiotic clearance of intracellular infections, HeLa epithelial cells were infected with each SL1344 strain. Infected HeLa were subjected to no antibiotic, 0.5 µg/mL TET, or 0.5 µg/mL TMP for 18 h of post infection. HeLa were subsequently lysed, and CFUs of intracellular SL1344 were determined. The surviving SL1344 in the presence of antibiotic were compared to the relative surviving *Salmonella* in the absence of antibiotic. Significant reductions in viable SL1344 were observed in the presence of TMP for both the three-gene ($P = 0.04$) and six-gene perturbation ($P = 0.03$) strains (Fig. 5f, g). This was also true of the six-gene perturbation strain's growth in presence of TET ($P = 0.008$). These results indicate that the targeted multiplexed CRISPRi constructs successfully potentiated intracellular antibiotic treatment, supporting the therapeutic viability of fitness-neutral gene perturbation treatments.

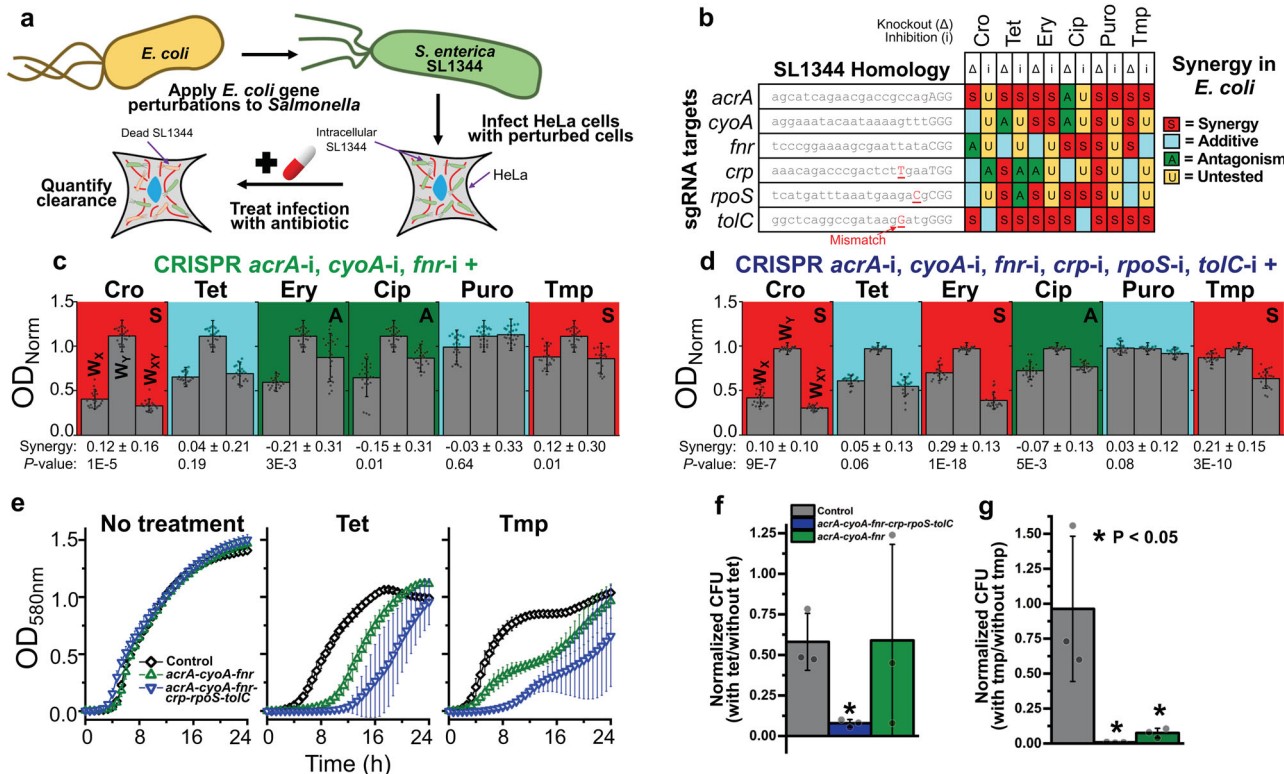

**Fig. 5 CRISPRi potentiation of antibiotic treatment of intracellular *Salmonella* infections. a** CRISPRi treatments that were demonstrated to be effective in *E. coli* and maintained significant homology to the genome of *Salmonella* were applied to *Salmonella* SL1344 cells. These perturbed SL1344 cells were used to infect HeLa epithelial cells to observe their ability to potentiate antibiotic treatment in a clinically relevant setting. **b** The exact 20 nt target sequences of six CRISPRi constructs are listed, with the native PAM (protospacer adjacent motif) sequence listed in capitals at the end of each sequence. Underlined red sequences indicate a mismatch in the sgRNA sequence with the native sequence of *Salmonella enterica* serovar Typhimurium SL1344. On the right is shown how these gene knockouts (Δ) or CRISPRi knockdowns (i) interacted with the corresponding antibiotic. **c, d** Two CRISPRi constructs targeting the genes with perfect homology (*acrA, cyoA,* and *fnr,* **c**) or all six genes (**d**) were created and screened for their ability to potentiate antibiotic treatment of SL1344. Growth of the control six *rfp* perturbation strain during exposure to the listed antibiotic, growth of the multiplexed CRISPRi strains, and growth of the multiplexed CRISPRi strains in the presence of the listed antibiotic were all normalized to the growth of the six *rfp* control strain without exposure to antibiotic, giving $W_X$, $W_Y$, and $W_{XY}$ respectively. Significant synergy or antagonism is indicated by a red background and the letter "S" or a green background with the letter "A", with blue representing additive interactions. Synergy values are listed below each graph with significance. Error bars represent standard deviation of 22 biological replicates and dark gray circles represent individual biological replicates. **e** Growth curves of CRISPRi SL1344 strains in the presence or absence of antibiotic treatment. Error bars represent standard deviation of at least five biological replicates. **f, g** Survival of CRISPRi SL1344 strains in intracellular HeLa infections after 18 h of 0.5 μg/mL TET (**f**) or 0.5 μg/mL TMP (**g**) treatment, relative to survival with no antibiotic treatment. *P* values are given in relation to the control strain. Error bars represent standard deviation of three biological replicates and two technical duplicates; gray circles represent individual biological replicates.

**PNA knockdown of gene expression potentiates antibiotic treatment of MDR clinical isolates**. To further explore the therapeutic potential of fitness neutral gene perturbations, we utilized an alternative gene expression knockdown approach based on PNA. The structure of PNA and DNA are similar, and the ability of PNA to bind to RNA has been well established[12]. When conjugated to CPPs, PNAs can readily cross bacterial membranes and enter the cell. When these PNAs enter the cell, they form tight bonds with complementary mRNA, preventing ribosome translation of these genes into proteins (Fig. 6a)[13]. This approach can be readily applied to a wide array of bacteria to induce gene expression knockdown in a plasmid-independent fashion.

We applied PNA knockdown of gene expression to four clinically isolated, MDR bacteria obtained from the University of Colorado Anschutz Medical Campus. Each strain has been previously sequenced and found to exhibit a wide range of antibiotic resistances[17,54–56]. This includes two strains of MDR *E. coli*, one of which exhibits a carbapenem-resistant

Enterobacteriaceae (CRE) phenotype, and two strains of *Klebsiella pnuemoniae* (KPN) producing either an extended spectrum β-lactamase (ESBL) or a New Delhi metallo-β-lactamase 1 (NDM-1). These strains have been found to survive a wide range of antibiotic concentrations significantly above the resistance breakpoint levels established by the Clinical & Laboratory Standards Institute (CLSI), including AMP, CIP, CM, TET, kanamycin (KAN), rifampicin, streptomycin, gentamicin (GEN), and clindamycin[8]. Additionally, we found each MDR strain to resist TMP levels well above the CLSI breakpoint of 2.0 μg/mL, while wildtype BW25113 exhibited sensitivity at 0.25 μg/mL (Fig. 6b and Fig. S10).

We chose to focus specifically on synergy with TMP, as we achieved the greatest success in engineering synergy in a fitness-neutral fashion with this antibiotic in our CRISPRi approach. We first screened the genomes of the four MDR isolates for homology with each of the six TMP related gene perturbations tested with CRISPRi. Low homology was found for *ydiV*, so we chose to exclude testing of this gene. The remaining five genes showed

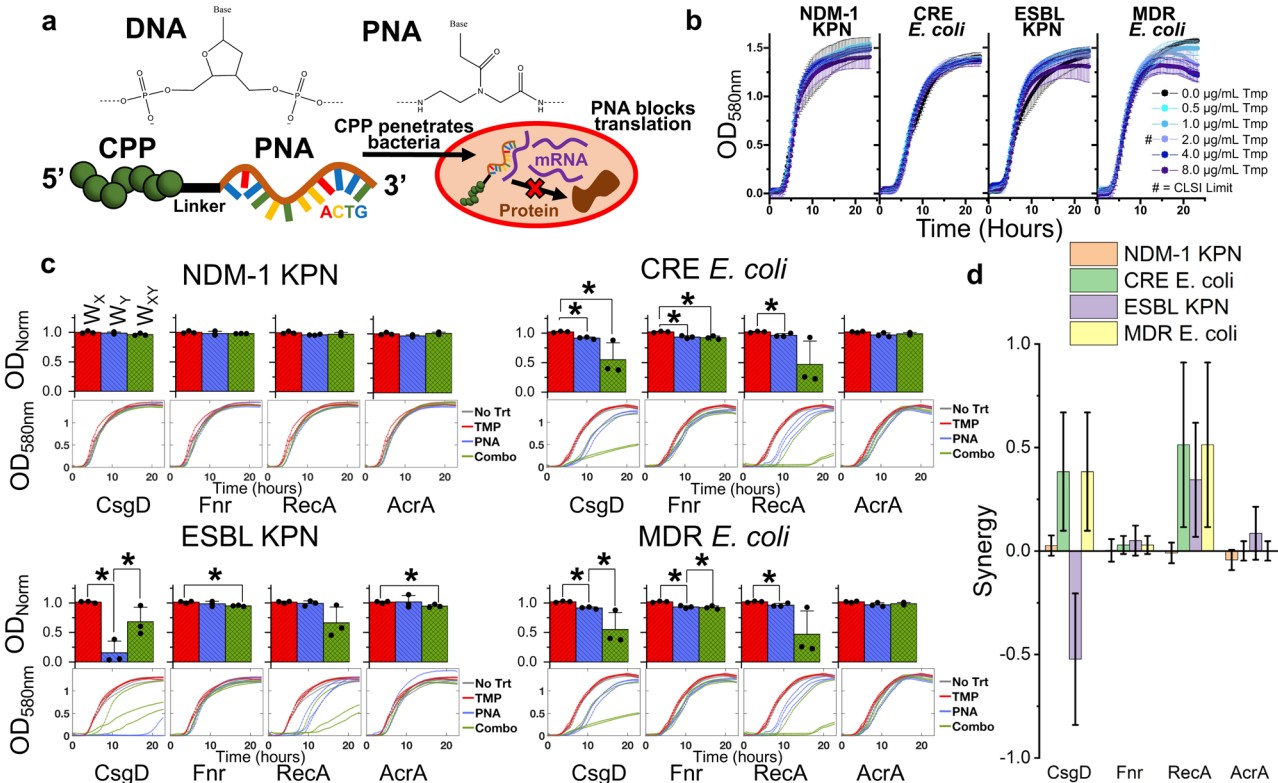

**Fig. 6 PNA gene knockdown treatment resensitizes MDR clinical isolates to antibiotics. a** Chemical structures of DNA and PNA show how the negatively charged phosphate backbone of DNA is replaced with a neutrally charged peptide backbone in PNA. These PNAs are conjugated to a CPP to enable penetration of bacterial membranes. Upon entry to the cell, PNAs complex with complementary mRNA to inhibit protein translation. **b** Resistance of MDR, clinically isolated bacteria to TMP above CLSI breakpoint levels of resistance, as demonstrated by growth curves unaffected by TMP concentration. Error bars represent standard deviation of four biological replicates. **c** MDR bacteria growth after exposure to 2 µg/mL TMP (red bars, $W_X$), 10 µM PNA (blue bars, $W_Y$), or both (green bars, $W_{XY}$). Bar plots (top) show growth in each condition normalized to blank wells and starting OD580 values and are subsequently normalized to the maximum growth in the absence of treatment. Growth curves (bottom) show OD580 values of individual biological replicates (3) over time with the minimum value subtracted. **d** Synergy values of PNA with TMP, grouped by specific PNA targets. Error bars represent standard deviation of biological triplicates and black circles represent individual biological replicates.

significant homology with the four MDR strains. However, we chose to exclude testing *tolC* as well, as multiple off-targets were found. Additionally, the TolC-AcrAB efflux pump has been targeted by similar expression interference techniques[25,57]. Excluding a *tolC* PNA minimizes redundancy, while allowing us to focus on other more novel targets. We designed 12 nucleotide (nt) long PNA molecules to inhibit expression of the remaining four genes (*acrA, csgD, fnr*, and *recA*) by targeting them to overlap the start codon of each gene's open reading frame. Target sequences are presented in Table S3, and any potential off-targets identified in a genome-wide screen are noted in Table S4. A control PNA targeting a nonsense sequence not present in any of the genomes was also designed. Testing the impact of this nonsense PNA on the growth of each MDR bacterial strain revealed that it had minimal impact on growth in the presence of TMP, indicating that PNA-CPP molecules have no inherent detrimental impact on MDR bacteria growth independent of the effects caused by targeted gene repression (Fig. S11).

We next examined the ability of the four targeted PNAs to synergize with TMP treatment in each of the four MDR bacterial strains (Fig. 6c, d). Again, TMP treatment alone demonstrated no effect on growth. Three PNAs exhibited impacts on MDR bacterial growth in the absence of TMP; the *recA* and *fnr* targeting PNA minimally reduced growth (<10% reduction) in both *E. coli* strains, while the *csgD* PNA minimally reduced

growth in both *E. coli* strains and significantly reduced growth (>10% reduction) in ESBL KPN.

For four out of the five instances where PNA showed a small (<10% reduction) impact on growth there is a greater than 50% reduction in growth for the combined PNA and antibiotic treatment condition as compared to PNA treatment alone for two out of the three biological replicates. Only one biological replicate showed greater than 50% reduction in growth for PNA targeting *recA* combined with antibiotic treatment in ESBL KPN. Due to the variability seen in these clinical isolates no combination showed statistically significant synergy at three biological replicates. The five cases with positive *S* values above 0.25 correspond with the clinical isolates that showed at least one biological replicate having greater than 50% reduction in growth in combination treatment as compared to PNA treatment alone.

While these findings substantiate the perturbation-drug synergy discovery pipeline outlined throughout this study, the large variability in synergy (caused by one replicate) suggests the potential for escape. Further research is necessary to understand the underlying mechanisms for this phenomenon.

## Discussion
Here we present a drug-discovery-pipeline for the identification of sequence-specific gene expression therapies that potentiate

antibiotic treatments without directly affecting bacterial fitness. We utilize a gene knockout library to reveal promising gene candidates and build upon these results by engineering CRISPRi and PNA therapies to facilitate targeted gene knockdown. Many of these gene expression treatments were shown to illicit no fitness cost when administered independently yet were still successful in potentiating antibiotic killing. Finally, we demonstrate that this approach is reproducible in an intracellular-infection context and is effective at resensitizing clinically isolated MDR bacteria to antibiotic treatment. Together, these experiments outline a pipeline for discovering perturbation-drug synergies applicable to therapeutic contexts. Notably, this pipeline could easily be expanded to explore more perturbations and drugs, to enhance the number of successful synergistic therapies discovered at the final stage.

While previous studies have employed the Keio library to explore the impacts of gene knockouts on antibiotic efficacy[24,58,59], these studies have not considered how identified gene–drug synergy can be therapeutically exploited to potentiate antibiotic efficacy, as we have done here. Likewise, previous studies examining how single gene disruption alters antibiotic fitness have lacked direct comparison to wild type controls, meaning that deleterious interactions cannot be defined as additive, synergistic, or antagonistic[24,58]. A lack of wild type performance data prevents other large-scale explorations of bacterial phenotype from being capable of typifying gene/drug interactions[22]. The work presented here, and similar efforts that use CRISPR-interference to perturb gene expression in the presence of stressful environments[11], are critical to explore the widespread potential that synergistic, antagonistic, or additive interactions may have. Another key consideration in this study is the concern placed towards designing fitness-neutral therapies. The field of antibiotic synergy research is well established, and numerous reports have emerged of a correlation between the application of synergistic antibiotics and an acceleration in the rate of antibiotic resistance evolution[60]. Recent studies indicate that during simultaneous application of synergistic antibiotics, mutations conferring resistance to one antibiotic are selected for at a greater rate than they would be if similar levels of that antibiotic were administered independently[61]. This is because the fitness deficit applied by the second synergistic antibiotic strengthens the selective advantage a resistant mutation provides by imposing an additional fitness benefit that would be absent in the case of application of one antibiotic. Crucially, this phenomenon should work in both directions; the synergy of antibiotic "A" provides an additional selective advantage for a resistance mutation for antibiotic "B" to emerge and vice-versa.

In our approach to developing fitness-neutral gene perturbations for inducing antibiotic synergy, the mutual selective advantage for the emergence of resistance mutations could theoretically be abated. A mutation providing resistance against the fitness-neutral gene perturbation has no selective pressure for propagating in the case of its independent application. Furthermore, in the case of co-application of the perturbation with an antibiotic, the selective advantage of a resistance mutation nullifying the impact of the perturbation will theoretically be less powerful, or indeed negligible, than a corresponding mutation for a fitness-impacting secondary antibiotic, ultimately decreasing the likelihood of such a mutation from emerging. In a clinical context, this would mean that fitness-neutral potentiating therapies could provide the enhanced killing caused by antibiotic synergy, while minimizing the potential for antibiotic resistance from emerging. Notably, the phenomenon of enhanced mutation rates caused by antibiotic synergy has only been considered between antibiotics that each exhibit a fitness impact on bacterial growth.

To our knowledge, no study has considered how replacing one of these antibiotics with a fitness-neutral treatment would alter the rate of antibiotic resistance from emerging. Further experiments utilizing the approach outlined in this study could explore this theory in future work.

The targeting of nonessential pathways to combat antibiotic resistance is an underexplored strategy in the literature, despite its potential for potentiating treatment without having an effect on pathogens on their own[62]. While essential genes interference therapies can be developed, the likelihood that their inherent deleterious effects on fitness will encourage the evolution of new resistance requires significant consideration. The potentiating CRISPRi and PNA knockdowns of gene expression provides a promising approach to enhance our ability to treat MDR bacteria in the clinic without selecting for further resistance. This strategy is supported by a similar approach in which interference of LexA activity was applied to reduce expression of *dinB*, *polB*, and *umuD*, resulted in significant potentiation of CIP and rifampicin treatment during a long-term treatment[63].

It is well known that the local environment can affect a therapy's efficacy[64–66]. While multiple environmental conditions including an infection model were explored in this study, follow up studies utilizing more thorough exploration in diverse environments, including in vivo mammalian models beyond the HeLa cell infections used in Fig. 5, will be required. Additionally, a number of our perturbations that initially appeared to induce no significant impact on fitness on their own, had clear fitness impacts when applied using PNAs in Fig. 6. This highlights why studies such as these will be important to understand the perturbations' effects in different environments, to establish strong clinical viability.

Finally, the pool of genes that can be explored for gene-drug synergies is vast and remains largely untapped. Here we explored only 30 of the reported 3985 nonessential genes of *E. coli*[18], many of which could hold promise as targets for fitness neutral potentiation of antibiotic therapy. Work in our lab has suggested that multiplexing gene perturbations can restrict the evolvability of bacteria. Through our approach, dubbed Controlled Hindrance of Adaptation of OrganismS or CHAOS, combinatorial gene expression knockdowns are introduced to disrupt adaptive pathways by eliciting negative epistatic interactions. This strategy has been demonstrated to significantly slow the increase in antibiotic MICs that bacterial populations could survive over time[47]. The data presented here suggests that the CHAOS approach might also facilitate resensitization of antibiotic resistant bacteria to treatment, while concurrently inhibiting the emergence of novel antibiotic resistance. Taken together, we are optimistic about the potential of gene/drug combination therapies to realize promising candidates with clinical relevance for combating antibiotic resistance.

## Methods

**Target gene selection**. We selected thirty genes to evaluate as potential targets for combination therapy. Many of these genes were chosen due to existing evidence of their association with stress response and/or adaptation processes. We previously quantified the behavior of certain SOS response (*recA*, *polB*, *dinB*, and *dam*), general stress response (*rpoS*, *hfq*, *cyoA*, and *mutS*) and mar regulon (*marA*, rob, soxS, acrA, and tolC) genes during adaptation[19]. Several of the genes selected here were also found to impact adaptive resistance in a transcriptome-level analysis of adapted versus unadapted strains (*fiu*, *tar*, *wzc*, *yjjZ* were differentially expressed while *ybjG*, *ydhY*, *ydiV*, and *yehS* were differentially variable)[20]. Finally, we looked upstream and selected transcriptional regulators that control the expression of these and other genes (*bglG*, *crp*, *csgD*, *flhC*, *flhD*, *fnr*, *fur*, *gadX*, and *phoP*). The selected genes represent diverse functionalities, including transport (*acrA*, *tolC*, and *fiu*), mutagenesis (*mutS*, *dam*, *polB*, *dinB*), motility (*tar*, *flhC*, *flhD*, and *ydiV*), general global regulation (*rpoS*, *marA*, *fnr*, *fur*, *gadX*, and others), and include a few targets with unknown function (*yjjZ* and *yehS*).

**Bacterial strains and culture**. All knockout strains used are from the Keio collection[18]. The parent strain (*Escherichia coli* BW25113) and individual gene knockouts were obtained from Yale's Coli Genetic Stock Center (http://cgsc.biology.yale.edu/index.php). *E. coli* NEB 10-β was used for cloning of all CRISPR plasmids used in this study. Experiments using plasmids were done by transforming these plasmids into *E. coli* BW25113. *Salmonella enterica* serovar Typhimurium SL1344 with genome-integrated Green Fluorescent Protein (GFP) was used for harboring CRISPR plasmids for intracellular infections.

Clinical isolates of multidrug resistant bacteria were obtained from the lab of Dr. Nancy Madinger at the University of Colorado Anschutz campus. This includes carbapenem-resistant Enterobacteriaceae (CRE) *E. coli*, another multidrug resistant *E. coli*, *Klebsiella pnuemoniae* harboring New Delhi metallo-β-lactamase 1 (NDM-1), *Klebsiella pnuemoniae* harboring extended-spectrum β-lactamase (ESBL)[54,55].

Unless otherwise noted, all experiments using these strains were performed in Lysogeny broth (LB). Experiments performed using M9 minimal media were supplemented with 2.0 mM MgSO$_4$, 0.1 mM CaCl$_2$, and 0.4% glucose. Experiments performed using cation-adjusted Mueller Hinton Broth (caMHB) contain MHB supplemented with magnesium and calcium ions. Keio collection strains were grown in the presence of 40 μg/mL kanamycin (KAN). Strains harboring CRISPR plasmids were grown in the presence of 100 μg/mL ampicillin (AMP) and 35 μg/mL chloramphenicol (CM) and supplemented with 50 ng/mL anhydrous tetracycline (aTc) for induction of dCas9 when appropriate. All liquid cultures were grown at 37 °C with 225 rpm shaking, and all plating was performed at 37 °C.

**Determining sub-minimum antibiotic concentrations**. AMP, ceftriaxone (CRO), gentamicin (GEN), KAN, puromycin (PURO), and sulfadimidine (SDI) were prepared with water as a solvent. Tetracycline (TET), erythromycin (ERY), and CM were suspended in 70%, 100%, and 100% ethanol, respectively. Ciprofloxacin (CIP) was prepared in water with HCl added drop by drop until the powder became soluble. Trimethoprim (TMP) was suspended in DMSO. All antibiotics were stored in aliquots at −20 °C.

BW25113 was plated from a glycerol stock and grown for 16 h. Three to five colonies were used to inoculate a 1 mL culture in M9 and grown for 16 h. Samples were subsequently normalized to OD$_{600}$ = 1. A 1:100 dilution was used to inoculate 50 μL cultures in M9 minimal media containing one of ten concentrations for each antibiotic (in two-fold increments) as well as controls without antibiotic. Optical density (OD) was monitored in a GENios plate reader (Tecan Group Ltd.) operating under Magellan software (version 7.2), with measurements taken every 20 min. The microplate reader was set to shake for 1000 s, with 10 s of shaking before measurement. The concentration selected for each antibiotic was that at which maximum OD was between 50–90% that of the control (Fig. S1).

To determine the antibiotic resistance of the MDR bacteria to TET and TMP, four individual colonies were inoculated into 3 mL caMHB and grown overnight for 16 h. Samples were then diluted 1:10,000 in fresh caMHB, of which 45 μL was aliquoted into a 384-well plate and supplemented with 5 μL of 10× antibiotic concentration of interest. Samples were then grown in a GENios plate reader for 24 h using the process described above.

**Characterizing gene knockout synergy with antibiotics**. BW25113 and single gene knockout mutants were plated from glycerol stocks. Colonies from each were used to inoculate 1 mL cultures in M9 minimal media with 0.4% glucose and grown for 16 h. Samples were then normalized to OD$_{600}$ = 1 (for Keio collection strains) or OD$_{580}$ = 1 (for all other strains) and diluted 1:100 into 50 μL cultures in M9 media containing either no antibiotic or the specified concentration of each antibiotic. Four biological replicates were included for each condition. OD was monitored in a Tecan GENios microplate reader as described above. The maximum OD$_{600}$ (for Keio strains) or OD$_{580}$ (for all other strains) achieved for each well was recorded and used for subsequent characterization of the nature of the interaction.

**Characterizing CRISPR gene knockdown synergy with antibiotics**. BW25113 harboring an RFP-targeting sgRNA was used as a nonsense wild type control for comparing the impact of perturbations, as this guide has no significant homology to the genomes of any strain used in this study. This strain, as well as single and multiplexed gene perturbation mutants were plated from glycerol stocks, and 20–22 individual colonies were used to inoculate 100 μL cultures in M9 minimal media with 0.4% glucose and AMP and CM selection. Cultures were grown for 16 h and diluted 1:100 into 100 μL cultures containing 50 ng/mL aTc and either no antibiotic or the specified concentration of each antibiotic. ODs were monitored in a Tecan GENios microplate reader as described above, and the maximum OD achieved for each well was recorded and used for subsequent characterization of the nature of the interaction. This same process was used for characterizing synergy of *Salmonella* strains harboring CRISPR perturbations, except M9 media was replaced with LB media.

**Fitness assay**. CRISPR strains harboring either six copies of RFP sgRNA perturbations, multiplexed TET or TMP synergistic gene perturbations, or a single RFP sgRNA perturbation and constitutively expressed mCherry were plated from glycerol stocks. Eight biological replicates of each strain were used to inoculate

200 μL LB cultures supplemented with AMP and CM and grown for 16 h. Samples were then diluted 1:100 into fresh media supplemented with AMP, CM, and 50 ng/mL aTc, and grown for another 24 h. Competition began by mixing 1 μL of the mCherry control strain with one μL of the competitor strain in 198 μL of the noted condition. Conditions always included LB supplemented with AMP and CM, three conditions with 50 ng/mL aTc, and another three conditions without aTc. Each condition was supplemented with either no additional antibiotic, 0.25 μg/mL TET, or 0.125 μg/mL TMP. Two microliter of each culture were used immediately to determine starting ratios of red to white cells. The remaining culture was grown for 24 h, diluted again 1:100 in fresh media, and grown for another 24 h, after which 2 μL was again used to determine ending ratios of red to white cells.

Ratios were determined by plating 50 μL of 1:10,000 and 1:100,000 on plain LB plates. Two plate images were taken with fluorescence activation at 540 nm, one with emission filtering at 590 nm and the other without, and these images were overlaid to facilitate colony counting. Colony counts were used to determine fitness values (ω) using the standard Malthusian fitness equation, using the formula $\omega = \ln(N_{E1} \times 100^2/N_{E0})/\ln(N_{C1} \times 100^2/N_{C0})$[67] where the variables are defined as follows: "N"—colony forming units (cfu), "E"—experimental strain, "C"—control strain, "1"—after exposure, and "0"—before exposure.

**Quantifying gene-antibiotic synergy**. The maximum values of OD$_{600}$ (for Keio collection strains) or OD$_{580}$ (for all other strains) reached in the presence or absence of antibiotic and gene knockouts/perturbations were then used to determine the degree of synergy. If $d_\Phi$ is the maximum OD of BW25113 wild type in media without antibiotic, $d_A$ is the OD$_{max}$ of the wild type with antibiotic treatment, $d_K$ is that of the mutant without antibiotic, and $d_{AK}$ is that of the mutant with antibiotic, then OD$_{max}$ can then be normalized as $W_X = d_A/d_\Phi$, $W_Y = d_K/d_\Phi$, and $W_{XY} = d_{AK}/d_\Phi$. Similarly, for gene perturbation experiments, if $d_\Psi$ is the maximum OD of the control perturbation strain in media without antibiotic, $d_X$ is the OD$_{max}$ of the control strain with antibiotic treatment, $d_Y$ is that of the perturbation without antibiotic, and $d_{XY}$ is that of the perturbation with antibiotic, then OD$_{max}$ can then be normalized as $W_X = d_X/d_\Psi$, $W_Y = d_Y/d_\Psi$, and $W_{XY} = d_{XY}/d_\Psi$. Furthermore, the degree of interaction synergy ($S$) was identified using the equation $S = W_X * W_Y - W_{XY}$.

**CRISPR plasmid and strain construction**. CRISPR expression was driven from two plasmids, one encoding for aTc-inducible dCas9 (Addgene plasmid 44249), and another encoding for constitutively expressed sgRNA targets derived from Addgene plasmid 44251. The latter plasmid was used to create new sgRNA target plasmids by replacing the RFP-targeting sequence with new gene sequences specific to the target of interest. Unique forward primers flanked with a SpeI restriction site and encoding the new target sequence and a common reverse primer flanked with ApaI was used to PCR amplify (Phusion, New England Biolabs) new DNA inserts, which were subsequently digested with CutSmart SpeI and ApaI (New England Biolabs) alongside 44251 backbone. Ligation of these pieces was performed using T4 DNA Ligase (Thermo Scientific), which were subsequently transformed into electrocompetent NEB 10-β. Plasmids from transformants were recovered using the Zyppy Plasmid Miniprep Kit (Zymo Research Corporation) and submitted for sanger sequencing to confirm successful insertion (GENEWIZ). These final plasmids were co-transformed alongside the dCas9 plasmid into BW25113 to create the final CRISPR individual perturbation strains. The specific sgRNA target sequences are presented in Table S1.

To create multiplexed gene perturbation strains, Gibson assembly was used to sequentially stich individual sgRNAs together. A common set of six forward and six reverse primers were used to amplify sgRNAs as Gibson fragments, beginning with stitching sgRNAs together in pairs. Once the paired sgRNA plasmids were confirmed, two of the three pairs for each set were stitched together using another round of Gibson assembly. Finally, once these four sgRNAs were confirmed, the final pair of sgRNAs were stitched together with this four-target sgRNA construct using a final round of Gibson assembly. Final plasmid sizes were confirmed, and then transformed into BW25113 or SL1344 for experiments. All Gibson reactions were performed at 50 °C for 1 h with T5 exonuclease (New England Biolabs), Phusion polymerase and Taq ligase (New England Biolabs). This process was used to create the multiplexed six-gene targeting sgRNA constructs.

**Growth assay of SL1344 CRISPRi strains**. To demonstrate growth curves, at least five biological replicates of each strain were inoculated from individual colonies into 200 μL of LB containing AMP and CM in a conical 96-well microplate and grown for 16 h. After initial growth, 2 μL of each culture was used to inoculate 198 μL of fresh media containing AMP, CM, and conditionally aTc or antibiotics as noted. These cultures were grown in a flat bottom 96-well microplate in a Tecan GENios microplate reader for 24 h as described above.

**STRING database analysis**. Data for the protein–protein interaction networks presented in Fig. S2 was collected from STRING version 10.5[48]. The acquired counts of network nodes, node degrees, edges, and the overall cluster coefficient of the protein network were extracted directly from the database. A minimum confidence score of 0.40 was used to collect information from experiments, databases, co-expression, co-occurrence, gene fusions, and gene neighborhoods. The

maximum number of interactions was set to 250 for the first shell and zero for the second shell.

**HeLa culture and infection**. HeLa human epithelial cells were recovered from freezer stocks in 10% dimethyl sulfoxide (DMSO, Sigma) and full growth media consisting of Dulbecco's modified Eagle medium (DMEM, Fisher Scientific), 10% fetal bovine serum (FBS, Advanced, Atlanta Biologics), and 50 units/mL penicillin-streptomycin (P/S; Fisher Scientific). A single freezer stock at passage number six was split into three separate tissue culture flasks maintained as separate biological replicates. Cells were maintained at 37 °C with 5% $CO_2$ and controlled humidity, and media was changed every two-three days with subculturing at 80% confluency with 0.25% trypsin (HyClone). HeLa were used for infection three passages after starting from a freezer stock, and all experiments were performed with HeLa between passages 8 and 12. At 24 h prior to infection, HeLa were trypsinized off of tissue culture flasks and seeded into a 96-well tissue culture treated plate (Fisher Scientific) at 10,000 cells/mL in 100 μL full growth media.

The night before infection, SL1344 was inoculated from plates into 3 mL LB with AMP and CM selection and grown for 16 h. Cultures were then diluted 1:10 in LB with AMP and CM and grown for another four hours, after which samples were washed three times with PBS and normalized to the same $OD_{580}$. HeLa cells were washed three times with Dulbecco's PBS (DPBS, Fisher Scientific) and incubated in DPBS containing SL1344 for 45 min at 37 °C with 5% $CO_2$ and controlled humidity. After 45 min of infection media was replaced with DMEM+ FBS+ 100 μg/mL GEN and incubated for another 75 min to remove extracellular bacteria. A final media replacement was done with full growth media (without penicillin–streptomycin and supplemented with AMP, CM, 50 ng/μL aTc, and 40 μg/mL GEN) in either the presence or absence of 0.5 μg/mL TET or 0.5 μg/mL TMP and incubated for 18 h. To perform CFU analysis, HeLa cells were washed with DPBS thrice and lysed with 30 μL of 0.1% triton X-100 for 15 min at room temperature, after which 270 μL of PBS was added to each well and plated on plain LB agar to determine CFUs.

**PNA design**. PNA sequences were designed to bind in a centered region of the mRNA AUG start codons for the genes of interest. These 12-mer sequences were synthesized using a semi-automated platform published previously[17], and they consist of a KFFKFFKFFK cell-penetrating-peptide (CPP) sequence on the N-terminus, followed by an "O-linker" sequence connecting the CPP to the 12-mer nucleoside sequences with a peptide backbone. These PNA sequences were optimized to exhibit minimal off-target effects in the BW25113 genome (REFSeq: GCF_000750555.1) using a custom program described below. As recA exhibited an off-target around the start codon of the ndk gene, this PNA was redesigned to bind within the first 12 nts of the recA gene, beginning with the start codon. PNA sequences were also analyzed for their ability to bind to the genome of each of the clinically isolated MDR bacteria. All PNAs were found to have at least one possible target sequence in each of the MDR bacteria. PNAs were purchased from PNA Bio, and their sequences are presented in Table S3.

Our PNA Finder bioinformatics toolbox was used to design the PNA sequences, as published previously[17]. Briefly, a custom Python 3.7 script[17] was used to extract the reverse complements of 12-mer nucleotide sequences centered on the mRNA AUG start codons (STC) for genes of interest. Homology was determined using the Bowtie 2 (version 2.3.5.1) short-read alignment tool[68], allowing for one nucleotide mismatch within the sequence alignment. The BEDTools (v2.25.0)[69] "intersect" function was used to identify alignments that overlapped with genome features, and a custom Python script was used to parse this data and calculate the alignments' proximities to gene STCs. Off-target or homology inhibition was defined as a sequence alignment overlapping the STC of a gene that the PNA was not specifically designed to inhibit. Thermodynamic considerations for PNA sequences were screened for using a custom Python script designed to search for potential solubility and self-interference issues. The former was addressed by looking for purine stretches greater than five bases, a purine content of greater than 50%, or a guanine-peptide content of greater than 35%. The latter was addressed by looking for self-complementary sequences of greater than five bases.

**PNA synergy experiments**. Three biological replicates of BW25113 and MDR strains were inoculated from individual colonies in 3 mL caMHB and grown overnight for 16 h. Strains were then diluted 1:10,000 in fresh caMHB and aliquoted in 45 μL to a 384-well plate, to which 5 μL of 100 μM PNA treatment of interest was added (final concentration of 10 μM). Samples were grown for 24 h in a GENios microplate reader as described above to track growth over time. Blank wells and minimum $OD_{580}$ values were subtracted. Maximum OD values were recorded for each condition and normalized to the maximum OD of the no treatment.

**Statistics and reproducibility**. To determine if a gene knockout–drug interaction was synergistic, additive, or antagonistic we followed the statistical analysis as described by Demidenko et al.[46]. Briefly, treatment independence was determined first by a two-sided t-test, after log transformation of the data. If the interaction was statistically significant ($P < 0.05$) then a one-sided t-test was used to determine if the interaction was synergistic ($P < 0.05$) or antagonistic ($P > 0.05$). Pearson correlation coefficients and their corresponding $P$ values were calculated using linear fits with no weighting (OriginPro 9.3.226 software). All other $P$ values reported

were calculated using a standard two-tailed type II student's t-test. Standard deviations were estimated using appropriate propagation of error formulas excluding covariance terms.

**Reporting summary**. Further information on research design is available in the Nature Research Reporting Summary linked to this article.

## Data availability

All data needed to evaluate the conclusions in the paper are present in the paper and/or the Supplementary Materials, including source data in Supplementary Data 1. Additional data available from authors upon request. Sequence data for the MDR clinical isolates used to analyze PNA homology have been deposited in GenBank with the accession codes WWEV00000000.1, MSDR00000000.1, WWEX00000000.1, and WWEY00000000.1 for MDR E. coli, CRE E. coli, ESBL KPN, and NDM-1 KPN, respectively.

## Code availability

The PNA Finder toolbox[70] is available at https://github.com/taunins/pna_finder and requires Python 3.7, Bowtie 2 (version 2.3.5.1)[68], SAMtools (1.9)[71], and BEDTools (v2.25.0)[69]. To run on a Windows operating system a Window-compatible bash shell is required, and Mac operating systems are not currently supported.

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

## Acknowledgements

We thank Nancy Madinger for providing the clinically isolated MDR bacteria used in this study. Thanks to Thomas Aunins for his assistance in screening the MDR libraries for PNA homology. This work was supported by the National Science Foundation Graduate Research Fellowship (Award #DGE1144083) to PBO, and DARPA Young Faculty Award [grant number D17AP00024], National Science Foundation [grant number MCB1714564], and University of Colorado start-up funds to A.C. The views, opinions, and/or findings contained in this article are those of the author and should not be interpreted as representing the official views or policies, either expressed or implied, of the Defense Advanced Research Projects Agency or the Department of Defense.

## Author contributions

P.B.O., K.E.E. and A.C. conceived of the study. K.E.E. designed and performed all Keio knockout experiments and quantified antibiotic concentrations for sub-MIC growth of E. coli. P.B.O. designed and constructed all CRISPRi strains and performed all CRISPRi and PNA growth experiments. K.A.E., J.K.C. and P.B.O. performed infection experiments. P.B.O. and T.R.A. designed PNAs. P.B.O., K.A.E. and K.E.E. wrote the paper. All authors have approved of the final manuscript.

## Competing interests

The authors declare no competing interests.
