## [Peer Review File · Communications Biology]

Reviewers' comments:

Reviewer #1 (Remarks to the Author):

To summarize: 1) 30 Keio mutants were grown in the presence of 9 antibiotics at concentrations below their MIC and a synergy score was calculated based on the Bliss independence model and terminal density data, 2) A network analysis was performed between protein interaction network and synergy scores, 3) CRISPRi was used to decrease expression of 6 of the genes of interest, and synergy was tested against 6 antibiotics, 4) CRISPRi's were multiplexed with 3 or 6 guide RNAs and their ability to synergize with 6 antibiotics in a Salmonella infection model was assessed, 5) peptide-nucleic acid conjugates were used to decrease expression of four genes in 4 MDR strains and test for synergy with trimethoprim. In addition, an emphasis was placed on finding gene knock-downs that do not impact growth on their own but increase the effects of antibiotics. My comments are below.

1) Figure 1, it was not clear why a 1 standard deviation above or below a synergy score of 0 was labeled as synergistic or antagonistic. The authors should have used the statistical test they mention in the methods in this figure to identify values that were significantly above or below 0, rather than just a 1 standard deviation threshold.

2) Figure 2, the magnitudes of the correlations in the network analysis are fairly low, and I don't think this section adds much to the study; if anything I think it is distracting and the authors should consider removing it to improve readability.

3) Figure 4 and other figures. I understand the synergy score is based on a terminal density, as one might use in an MIC; however, the differences are much more stark and convincing when portrayed as growth curves. Just looking at the bar charts throughout it seems that most of the effects discussed are minor, but looking at some of the growth curves that correspond to those bar charts makes it much more obvious there is something there. I think the authors do themselves a disservice by presenting the data in the way they do. If terminal density is the key variable, it makes sense to use the endpoint, but if bacterial load over the entire time period is more important, than a different metric is warranted. I for one think the authors' claims are much more convincing when the entire time period is considered.

4) I think the authors should be applauded for the extrapolation to a salmonella infection model and MDR strains.

5) I think that the main criticism that can be rendered (other than that the effects seem to be minor the way the authors are presenting them, though I am guessing that they are not minor) is that the mechanisms of interactions are not explored at all, at least for the ones that aren't known. Increasing antibiotic efficacy by inhibiting efflux pumps or recA when you have antibiotic that damages DNA is not new information; however, some interactions like the ones with fnr could have been explored. It appears that the authors intent was not to go into mechanistic detail, since it reads as it is establishing the methodology. As far as this reader is concerned, I believe the authors could identify gene targets to synergize with a given antibiotic using their techniques, but why they synergize, not so much.

Reviewer #2 (Remarks to the Author):

This manuscript examines the impact of fitness neutral (in vitro fitness neutral) KO or silencing on antibiotic efficacy. The premise of the paper is very interesting and the approach of identifying synergistic targets useful, albeit not novel. The experiments are well thought out with the appropriate controls.

This is a re-review of a manuscript and may of the points raised by the original reviewers have been addressed except for point 2 from reviewer 1

"Many of the reported gene-antibiotic interactions are weak and remain doubtful. The synergy

criterion that is used seems to score cases as synergistic, where this does not match with a biologically relevant effect. In some instances, synergy is scored even when no or only very small effects on growth were observed”.

Reviewer 1 assumes this is due to a +ve impact of the CRISPRi perturbations on growth, however, the lack of biologically relevant effect is also clear in the non CRISPRi KO strains in Figure 1. The paper identifies (in figure 1) 114 synergistic reactions as calculated using the BLISS independence model. This model is a commonly used model to identify synergies between drug treatment. However, there is evidence in the literature that it is prone to false +ves as it does not take into account the variability of responses (<https://pubmed.ncbi.nlm.nih.gov/24492921/>). When looking at the data I am concerned that there are many false +ves (Figure 1, Figure 3, Figure 4). While there are some clear examples of synergy (e.g. Amp and Δ acrA) the impact of the deletion/inhibition in many of the combination does not appear to have an impact, despite the model calling synergy. One example being Δ acrA and CRO being one example of many.

Only a very small subset of the interactions have been explored further using growth curves. Therefore it is very hard to assess the reliability of the interaction data for Figure 1. Is it possible to explore the use of new models for synergy calls? How does this impact the data? The propensity for false +ve calling should at least be discussed.

Supplementary information has been provided to provide evidence for the CRISPRi perturbations, however it is still challenging to assess if this is truly synergy and so it would be over claiming to call these interactions synergistic – as they are referred to in the paper. Particularly when rpoS-i and csgD-i and Tet (Supplementary Figure 5) have a very similar responses in the growth curve analysis yet have been classified using the BLISS model as synergistic (csgD-i) and antagonistic (rpoS-i) (Figure 3- main text - opposite end of the spectrum in terms of interactions. This highlights even more the errors of the BLISS model in calling synergy for this dataset. Other revisions and comments have been addressed by the revision.

Reviewer #1 (Remarks to the Author):

To summarize: 1) 30 Keio mutants were grown in the presence of 9 antibiotics at concentrations below their MIC and a synergy score was calculated based on the Bliss independence model and terminal density data, 2) A network analysis was performed between protein interaction network and synergy scores, 3) CRISPRi was used to decrease expression of 6 of the genes of interest, and synergy was tested against 6 antibiotics, 4) CRISPRi's were multiplexed with 3 or 6 guide RNAs and their ability to synergize with 6 antibiotics in a Salmonella infection model was assessed, 5) peptide-nucleic acid conjugates were used to decrease expression of four genes in 4 MDR strains and test for synergy with trimethoprim. In addition, an emphasis was placed on finding gene knock-downs that do not impact growth on their own but increase the effects of antibiotics. My comments are below.

We thank the reviewer for their comments and have updated the manuscript to address their concerns as discussed below.

Specific Comments:

1. Figure 1, it was not clear why a 1 standard deviation above or below a synergy score of 0 was labeled as synergistic or antagonistic. The authors should have used the statistical test they mention in the methods in this figure to identify values that were significantly above or below 0, rather than just a 1 standard deviation threshold.

We thank the reviewer for their suggestion and have updated our statistical analysis of synergy using a method by Demidenko et al. (*PLoS One*, 2019) where, after log transformation of the data, a two-sided *t*-test was used to determine drug independence and then a one-sided *t*-test was used to determine synergism or antagonism. Where the two-sided *t*-test evaluates if the synergy value is statistically different from additive interaction and the two-sided *t*-test clarifies if there is interaction if it is antagonistic or synergistic.

2. Figure 2, the magnitudes of the correlations in the network analysis are fairly low, and I don't think this section adds much to the study; if anything I think it is distracting and the authors should consider removing it to improve readability.

We thank the reviewer for their comment and understand their concern. We agree that the analysis is not critical to the overall scope of the work and have therefore moved it to supplemental. We kept the analysis in supplemental because while the Pearson's coefficient is fairly low ($r = 0.168$) we believe that it adds to the strategy described in this paper for identifying genetic perturbations that are neutral by themselves but have significant interaction when paired with an antibiotic. Further we believe that analysis of the STRING network of more genetic and antibiotic combinations could provide more information into what genetic and antibiotic combinations would be good candidates for synergistic treatments.

3. Figure 4 and other figures. I understand the synergy score is based on a terminal density, as one might use in an MIC; however, the differences are much more stark and convincing when portrayed as growth curves. Just looking at the bar charts throughout it seems that most of the effects discussed are minor, but looking at some of the growth curves that correspond to those bar charts makes it much more obvious there is something there. I think

the authors do themselves a disservice by presenting the data in the way they do. If terminal density is the key variable, it makes sense to use the endpoint, but if bacterial load over the entire time period is more important, than a different metric is warranted. I for one think the authors' claims are much more convincing when the entire time period is considered.

We agree with the reviewer and as such have added growth curves to figures where possible. Figure 3, 4, 5, and 6 include growth curves so that the significance is more apparent. Unfortunately, the amount of data that is shown in Figure 1 made it difficult to show legible growth curves so we chose not to add growth curves to that figure.

4. I think the authors should be applauded for the extrapolation to a salmonella infection model and MDR strains.

We thank the reviewer for their comment and appreciate the statement.

5. I think that the main criticism that can be rendered (other than that the effects seem to be minor the way the authors are presenting them, though I am guessing that they are not minor) is that the mechanisms of interactions are not explored at all, at least for the ones that aren't known. Increasing antibiotic efficacy by inhibiting efflux pumps or recA when you have antibiotic that damages DNA is not new information; however, some interactions like the ones with fnr could have been explored. It appears that the authors intent was not to go into mechanistic detail, since it reads as it is establishing the methodology. As far as this reader is concerned, I believe the authors could identify gene targets to synergize with a given antibiotic using their techniques, but why they synergize, not so much.

We thank the reviewer for their comment and agree that the intent of our paper was to highlight a strategy for determining synergistic interactions, going from a simple and large matrix of data and narrowing down the pool of candidates. Here we specifically wanted to look at interactions that were more fitness neutral alone yet synergized in combination with an antibiotic, without mechanistic foresight. We agree that understanding the mechanisms behind why certain pairs synergized well across the different experimental setups is important but is beyond the scope of this paper.

Reviewer #2 (Remarks to the Author):

This manuscript examines the impact of fitness neutral (in vitro fitness neutral) KOs or silencing on antibiotic efficacy. The premise of the paper is very interesting and the approach of identifying synergistic targets useful, albeit not novel. The experiments are well thought out with the appropriate controls.

Specific Comments:

1. This is a re-review of a manuscript and may of the points raised by the original reviewers have been addressed except for point 2 from reviewer 1

“Many of the reported gene-antibiotic interactions are weak and remain doubtful. The synergy criterion that is used seems to score cases as synergistic, where this does not match with a biologically relevant effect. In some instances, synergy is scored even when no or only very small effects on growth were observed”.

Reviewer 1 assumes this is due to a +ve impact of the CRISPRi perturbations on growth, however, the lack of biologically relevant effect is also clear in the non CRISPRi KO strains in Figure 1. The paper identifies (in figure 1) 114 synergistic reactions as calculated using the BLISS independence model. This model is a commonly used model to identify synergies between drug treatment. However, there is evidence in the literature that it is prone to false +ves as it does not take into account the variability of responses (<https://pubmed.ncbi.nlm.nih.gov/24492921/>). When looking at the data I am concerned that there are many false +ves (Figure 1, Figure 3, Figure 4). While there are some clear examples of synergy (e.g. Amp and Δ acrA) the impact of the deletion/inhibition in many of the combination does not appear to have an impact, despite the model calling synergy. One example being Δ acrA and CRO being one example of many.

We thank the reviewer for their feedback and careful review which we used to improve the manuscript. In response to the reviewers concerns we have updated our definition of synergy using the statistical calculation describe by Demidenko et al. (*PLoS One*, 2019) where, after log transformation of the data, a two-sided *t*-test was used to determine drug independence and then a one-sided *t*-test was used to determine synergism or antagonism. This more rigorous statistical definition resulted in some previously interactions that were classified as synergistic to be reclassified as antagonistic and the Figures have been updated to reflect this. This metric is a more rigorous statistical definition of synergy, but it does not showcase the degree of synergy of each combination. As the reviewer noted there are some instances of strong synergy (Amp and Δ acrA) and weaker synergy (CRO and Δ acrA) which is why we always try and note the actual synergy value (as in Figures 3, 4, & 5) or graph the synergy values (as in Fig 2c which are the synergy values from Fig 2b).

2. Only a very small subset of the interactions have been explored further using growth curves. Therefore it is very hard to assess the reliability of the interaction data for Figure 1. Is it possible to explore the use of new models for synergy calls? How does this impact the data? The propensity for false +ve calling should at least be discussed.

We thank the reviewer for their feedback and have looked into alternative definitions of synergy including Loewe Additivity (LOEWE), CI, Highest Single agent (HAS), Zero

interaction potency (ZIP), etc. We found that the Bliss independence model is an effective method of analyzing synergy especially when looking at a large data set of individual combinations and looking at end point analysis (Meyer 2020 and Fouquier 2015). We believe that the more rigorous statistical analysis of synergy that we have implemented should account for any false positives seen in the data. In addition we have added the growth curves corresponding to the bar graphs in Figure 3 to demonstrate the reliability of the data.

*3. Supplementary information has been provided to provide evidence for the CRISPRi perturbations, however it is still challenging to assess if this is truly synergy and so it would be over claiming to call these interactions synergistic – as they are referred to in the paper. Particularly when *rpoS-i* and *csgD-i* and Tet (Supplementary Figure 5) have a very similar responses in the growth curve analysis yet have been classified using the BLISS model as synergistic (*csgD-i*) and antagonistic (*rpoS-i*) (Figure 3- main text - opposite end of the spectrum in terms of interactions. This highlights even more the errors of the BLISS model in calling synergy for this dataset.*

Other revisions and comments have been addressed by the revision.

We thank the reviewer for their feedback and note that the supplemental Figures S3-S8 are replicate experiments of the combinations seen in Figure 3. The data in Figure 3 was used to determine whether the CRISPRi and antibiotic combinations were synergistic and are the results of 20-22 biological replicates cultured in M9 media (similar to Figures 1). Supplemental Figures S3-S8 were replicate experiments of Figure 3 done in LB media using only 4 biological replicates and used as a comparison prior to determining which gene targets to move forward with for multiplexed experiments (Figure 4).